# Reinforcement Learning with Inverse Rewards for World Model Post-training

## Abstract

World models simulate dynamic environments, enabling agents to interact with diverse input modalities. Although recent advances have improved the visual quality and temporal consistency of video world models, their ability of accurately modeling human-specified actions remains underexplored. Reinforcement learning presents a promising approach for directly improving the suboptimal action-following capability of pre-trained models, assuming that an appropriate reward function can be defined. However, transferring reinforcement learning post-training methods to world model is impractical due to the prohibitive cost of large-scale preference annotations and the infeasibility of constructing rule-based video verifiers. To address this gap, we propose Reinforcement Learning with Inverse Rewards (RLIR), a post-training framework that derives verifiable reward signals by recovering input actions from generated videos using an Inverse Dynamics Model. By mapping high-dimensional video modality to a low-dimensional action space, RLIR provides an objective and verifiable reward for optimization via Group Relative Policy Optimization. Experiments across autoregressive and diffusion paradigms demonstrate 5–10% gains in action-following, up to 10% improvements in visual quality, and higher human preference scores, establishing RLIR as the first post-training method specifically designed to enhance action-following in video world models.

## 1 Introduction

World models aim to simulate dynamic environments, enabling intelligent agents to effectively interact with various input modalities such as robot actions, camera poses, or keyboard commands (Ha & Schmidhuber, 2018). Building on recent advances in generative modeling (Ho et al., 2020; Rombach et al., 2022) and large-scale video datasets, contemporary video world models achieve substantial improvements in both fidelity and diversity of synthesized visual environments through training action-conditioned video generation models (Hu et al., 2023; Guo et al., 2025b; Bar et al., 2025).

To function as reliable simulators, world models must satisfy three key requirements: producing high-fidelity visual content, maintaining long-horizon temporal consistency, and accurately following human-specified actions. While extensive research has addressed the first two challenges (Google, 2025), with approaches such as extending context windows (Liu et al., 2024; Zhang & Agrawala, 2025; Gu et al., 2025) and incorporating 3D priors (Wu et al., 2025c;a; Xiao et al., 2025), the problem of accurate action-following remains comparatively underexplored (Tot et al., 2025), despite its central role in controllable and interactive world modeling.

In natural language processing, reinforcement learning-based post-training has proven highly effective for aligning large language models with human preferences (Ouyang et al., 2022) and for enhancing reasoning capabilities (Wen et al., 2025). These successes suggest reinforcement post-training as a promising direction for video world models. However, direct transfer of existing techniques faces critical obstacles: (1) collecting human preference annotations at scale for video data is prohibitively expensive and prone to bias, and (2) while approaches such as RLVR (Wen et al., 2025) mitigate this issue by leveraging faithful, rule-based rewards and often achieve strong performance, their applicability remains limited to narrow domains (e.g., coding and mathematics). In particular, designing rule-based verifiers to reliably assess the quality of generated video is generally infeasible.

To overcome these challenges, we introduce Reinforcement Learning with Inverse Rewards (RLIR), a post-training framework for world models. The core idea is that, *rather than evaluating model output directly in the high-dimensional video space, RLIR derives reward signals in the low-dimensional input space (e.g., actions) by employing an inverse model that predicts conditioning actions from the generated videos*. Within our post training framework, we begin by employing either autoregressive or diffusion models to generate video sequences conditioned on input actions. An Inverse Dynamics Model (IDM) is then utilized to translate actions back from the generated videos. Given access to the ground-truth input actions, we can compare the inferred actions with the original input actions on a per-frame basis, thereby obtaining a verifiable reward signal. The reward increases with the degree of alignment between the inferred and ground-truth actions. Finally, we adopt the Group Relative Policy Optimization (GRPO) algorithm (Shao et al., 2024) to optimize the world model according to the relative advantages of the generated sequences.

Our approach is grounded in the key insight that, although multiple valid videos may correspond to the same action sequence, all high-quality generations must faithfully encode the input actions. Deviations such as temporal inconsistencies or visual artifacts reduce IDM accuracy, thereby naturally penalizing inferior outputs. Compared with human preference-based rewards, this action-consistency signal offers a more objective, scalable, and low-bias criterion for post-training world models.

We evaluate RLIR on interactive game generation domain across both autoregressive (next-token prediction) and diffusion world models. The results corroborate our key insight, demonstrating consistent improvements in action-following accuracy and visual quality across different generative paradigms. In summary, our contributions are threefold:

**i)** We introduce Reinforcement Learning with Inverse Rewards, a post-training paradigm that employs an inverse model to map inherently unverifiable video outputs into a verifiable, low-dimensional action sequence, thereby enabling reinforcement post-training to video world models.

**ii)** We leverage RLIR to improve action-following ability in world models, demonstrating its remarkable effectiveness across both autoregressive and diffusion paradigms. To the best of our knowledge, this is the first post-training method specifically designed to improve action-following ability.

**iii)** Experiments on both generative paradigms show consistent 5%-10% gains on action-following metrics and up to a 10% improvement in visual quality, with superior human preference scores.

## 2 RELATED WORK

### 2.1 WORLD MODEL

World models (Ha & Schmidhuber, 2018; Team et al., 2025; Zhou et al., 2024; Mao et al., 2025; Agarwal et al., 2025) are generative systems that enable agents or humans to effectively interact with dynamic environments. In the gaming domain, numerous studies (Guo et al., 2025b; Bruce et al., 2024; Yu et al., 2025a; Kanervisto et al., 2025) simulate interactive video games as well as real-world exploration, further extending the controllability and scalability of world models. Prior work has emphasized visual quality (Google, 2025) and long-horizon physical consistency (Xiao et al., 2025; Wu et al., 2025c) in world models, yet the issue of inaccurate action-following remains unaddressed. Our paper focuses on improving the action-following capability of world models.

### 2.2 REINFORCEMENT LEARNING FOR GENERATIVE MODELS

Reinforcement learning has emerged as a critical paradigm for post-training to better align with human preferences or task-specific objectives. DeepSeek-R1 (Guo et al., 2025a) introduces verifiable rewards and uses group relative policy optimization (GRPO) as its training method, which is more memory efficient by removing the need for a value network. Recently, GRPO-style methods (Liu et al., 2025a; Xue et al., 2025; Li et al., 2025) have progressed rapidly in generative models. However, their reward functions primarily rely on metrics such as aesthetics (Schuhmann et al., 2022), text-image alignment (Radford et al., 2021), or human preference scores (Wu et al., 2023), which are constrained by the accuracy and biases of the reward models and often result in suboptimal performance. In our work, we use action accuracy as a reward, an objective criterion that can be precisely measured

via an Inverse Dynamics Model, which is the first post-training method designed for improving action-following in video world models.

## 3 PRELIMINARIES

### 3.1 INVERSE DYNAMICS MODEL

Given a trajectory of $T$ observations $o_t : t \in [1...T]$, an Inverse Dynamics Model (IDM) estimates the action that transitions $o_t$ to $o_{t+1}$; formally, it models $p_{\text{IDM}}(a_t|o_t, o_{t+1})$. The IDM is trained on a contractor-labeled dataset by minimizing the negative log-likelihood of the ground-truth action at time $t$ given $(o_t, o_{t+1})$. Since the model leverages information from all video frames (including both past and future observations) to infer the current action, and given that the action space is substantially lower-dimensional than the raw video space, the IDM can achieve accurate prediction even with a limited amount of labeled data. The effectiveness of IDMs has been extensively validated in various domains, including robotic manipulation (Du et al., 2023; Black et al., 2023; Tan et al., 2025), game environments (Baker et al., 2022), and 3D geometric perception (Huang et al., 2025).

A well-trained Inverse Dynamics Model exhibits high sensitivity to minor visual artifacts and subtle variations of actions. In Figure 1 (left), we manually retouch only the cracks of the trunk to emulate a localized failure in world model generation. The IDM detects the inconsistency and consequently outputs an incorrect action prediction. As shown on the right, the IDM can also reliably discriminate between actions such as 'forward' and 'sprint', even when the visual differences are minimal. Prior work (Baker et al., 2022) further demonstrates the effectiveness of IDM in the Minecraft environment, reporting 90.6% accuracy on keypress prediction and an $R^2$ of 0.97 for mouse movement regression.

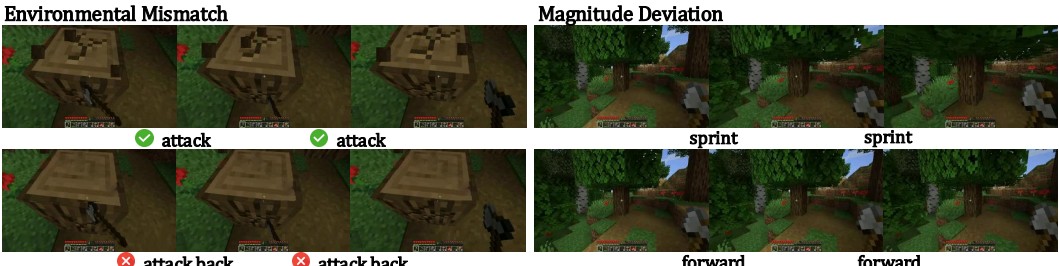

Figure 1: Inverse Dynamics Model (IDM) is highly sensitive to subtle environmental changes and action magnitudes. **(left)** The IDM flags the failure to produce cracks on the trunk as the action 'attack,back' rather than the ground-truth 'attack' and therefore labels it as a negative sample. **(right)** The IDM detects even subtle differences in action magnitudes (e.g., 'forward' and 'sprint').

### 3.2 GROUP RELATIVE POLICY OPTIMIZATION

Group Relative Policy Optimization (GRPO) (Shao et al., 2024; Guo et al., 2025a) is originally developed for post-training LLMs with reinforcement learning. Compared to Proximal Policy Optimization (Schulman et al., 2017), GRPO dispenses with a value function and estimates advantages in a group-relative manner. Specifically, given a question $q$, it samples a set of $G$ responses $\{o_i\}_{i=1}^{G}$ from the behavior policy $p_{\theta_{\text{old}}}$, and computes the advantage of each response by normalizing its reward $R_i$ within the group:

$$\hat{A}_{i,t} = \frac{R_i - \text{mean}(\{R_i\}_{i=1}^{G})}{\text{std}(\{R_i\}_{i=1}^{G})} \tag{1}$$

Similar to PPO, GRPO uses a clipped objective with a KL divergence (Shlens, 2014) penalty:

$$\mathcal{J}_{\text{GRPO}}(\theta) = \mathbb{E}_{q \sim \mathcal{D}, \{o_i\}_{i=1}^{G} \sim p_{\theta_{\text{old}}}(\cdot|q)}$$

$$= \left[ \frac{1}{G} \sum_{i=1}^{G} \frac{1}{|o_i|} \sum_{t=1}^{|o_i|} \left( \min \left( \frac{p_\theta^{i,t}}{p_{\theta_{\text{old}}}^{i,t}} \hat{A}_{i,t}, \text{clip} \left( \frac{p_\theta^{i,t}}{p_{\theta_{\text{old}}}^{i,t}}, 1-\varepsilon, 1+\varepsilon \right) \hat{A}_{i,t} \right) - \beta D_{\text{KL}} \left[ p_\theta || p_{\text{ref}} \right] \right) \right], \tag{2}$$

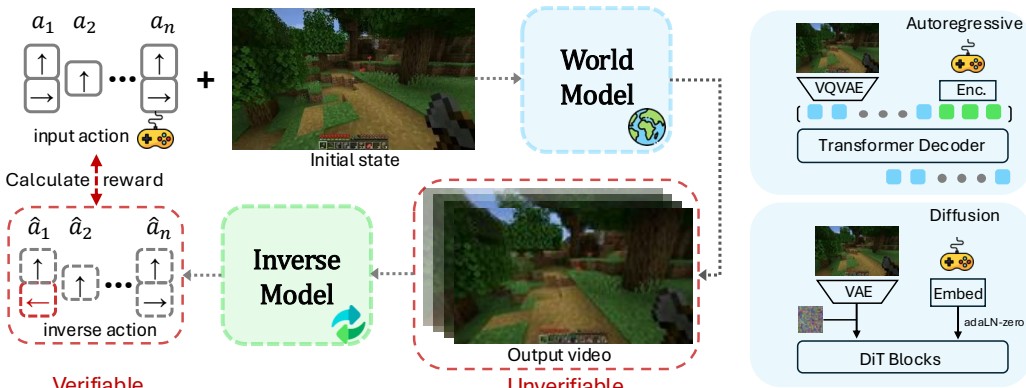

Figure 2: **Overview of RLIR.** Given the input actions, the world model generates video sequences conditioned on the input actions. An Inverse Dynamics Model (IDM) is then utilized to derive verifiable reward signals by recovering input actions from generated videos. We adopt Group Relative Policy Optimization (GRPO) to optimize the world model according to the alignment between the inferred and ground-truth input actions. We validate RLIR on both autoregressive and diffusion world models; architectures are shown on the right.

where $p_\theta^{i,t}$ denotes $p_\theta(o_{i,t} \mid q, o_{i,<t})$ for simplicity. Numerous recent works (Yu et al., 2025b; Zheng et al., 2025; Shrivastava et al., 2025) optimize GRPO for algorithmic efficiency or performance. For simplicity, we adopt the vanilla GRPO algorithm in this paper.

# 4 METHOD

In this section, we describe our method in detail. We first briefly introduce the problem and motivations. Then in Section 4.1, we describe how an Inverse Dynamics Model (IDM) is used as the reward model in Reinforcement Learning with Inverse Reward (RLIR). Sections 4.2 and 4.3 demonstrate the application of RLIR to two representative classes of world models, namely autoregressive world model and diffusion world model, respectively.

**Problem Definition**   To provide a simplified description of world models, we denote the model-generated frames as $\hat{x}$. Given an initial state $x_0$ and an action sequence $a_1, \ldots, a_n$, the world model generates each frame $\hat{x}_i$ conditioned on the initial state $x_0$, the previously generated frames $\hat{x}_1, \ldots, \hat{x}_{i-1}$ and the corresponding actions $a_1, \ldots, a_i$.

**Motivation**   To ensure that world models accurately follow human-specified actions, we focus on enhancing their action-following capability. Our approach is based on the insight that if the generated video frames faithfully reflect the input actions, then these actions can be reliably recovered from the generated frames. Guided by this insight, we initialize with a pretrained video world model and incorporate an IDM as a reward function to enhance action alignment through reinforcement learning.

## 4.1 IDM AS REWARD MODEL

Research on applying reinforcement learning to world models remains relatively limited. Existing approaches (Liu et al., 2025b; Wu et al., 2023) primarily assess visual quality, but they cannot reliably determine whether the intended actions have been correctly executed at the frame level, and they are susceptible to bias. Assigning rewards becomes substantially easier if we map video back to the action space and evaluate rewards in the action space. Specifically, while a world model maps an action sequence to video, by projecting the generated video back into the action space and comparing it to the original actions, we obtain a direct measure of the model's action-following fidelity. This video-to-action mapping can be implemented simply and accurately using an IDM. Formally, for each generated trajectory $T_j = [x_0, \hat{x}_1, \ldots, \hat{x}_n]$, there exists a corresponding ground truth action sequence $a_1, \ldots, a_n$. We employ a well-trained IDM that takes the generated trajectory $T_j$ as input and predicts the actions $\hat{a}_1, \ldots, \hat{a}_n$. Given that the IDM has been thoroughly trained to predict actions

with high precision, any discrepancy between the predicted actions $\hat{a}_i$ and the ground truth actions $a_i$ can be attributed to errors in the world model. Our reward function can thus be formalized as follows:

$$R_{T_j} = \frac{1}{n} \sum_{i=1}^{n} r(\hat{a}_i, a_i), \qquad r(\hat{a}_i, a_i) \triangleq \begin{cases} 1, & \text{if } \hat{a}_i = a_i, \\ 0, & \text{otherwise.} \end{cases} \tag{3}$$

Unlike previous reward models for video generation, IDM is trained exclusively on videos with ground-truth action annotations, introducing no additional bias and yielding precise action estimates that translate directly into a verifiable reward signal.

## 4.2 AUTOREGRESSIVE WORLD MODEL

An autoregressive world model generates a video sequence by iteratively predicting the next visual token in the sequence. We utilize the pretrained MineWorld (Guo et al., 2025b) as our baseline model. MineWorld employs a visual-action autoregressive Transformer that takes pairs of game scenes and corresponding actions as input and generates subsequent scenes conditioned on the actions. The inputs comprise two modalities: gameplay videos represented as a sequence of states $x_i$ and actions $a_i$ captured from mouse and keyboard events. For each state-action pair $(x_i, a_i)$, a VQ-VAE (Van Den Oord et al., 2017) tokenizer encodes $x_i$ into a sequence of quantized codes $t$, and an action tokenizer encodes $a_i$ into a flat sequence of discrete tokens separately. The tokenized data are structured as follows:

$$(t_0^{x_i}, \cdots, t_n^{x_i}, [\text{aBOS}], t_0^{a_i}, \cdots, t_8^{a_i}, [\text{aEOS}]). \tag{4}$$

The Transformer architecture follows LLaMA (Grattafiori et al., 2024) and treats tokens that represent game states and actions equally. The model is trained with next-token prediction to learn rich representations of game states and to model dependencies between states and actions.

Our post-training method is based on Group Relative Policy Optimization. In MineWorld, the rollout sequence comprises visual tokens and action tokens, and the latter are derived from external inputs. Optimizing visual tokens improves action-following ability and generative performance, but applying the same optimization to action tokens can induce undesirable training dynamics. During training, we address this challenge by implementing loss masking for action tokens, effectively disregarding the loss associated with these tokens. This ensures that the policy-gradient objective is computed solely on tokens generated by the world model, excluding action tokens from optimization.

## 4.3 DIFFUSION WORLD MODEL

In contrast to the autoregressive world model, the diffusion world model leverages Diffusion Forcing (Chen et al., 2024) to generate videos by autoregressively denoising future frames. We use the pretrained NFD (Cheng et al., 2025) as our baseline model. NFD uses diffusion Transformer blocks with block-wise causal attention. During inference, it can perform causal sampling across frames while applying parallel diffusion denoising to all visual tokens within each frame. NFD uses an image-level tokenizer to convert each frame into a sequence of tokens in a continuous space. For action processing, it uses a linear layer to map actions to vector embeddings, and AdaLN-Zero (Peebles & Xie, 2023) conditioning injects action information into the model.

Inspired by prior work (Xue et al., 2025; Liu et al., 2025a), we convert the deterministic Flow-ODE used in NFD into an equivalent SDE whose marginal probability density matches that of the original model at all timesteps. Within the diffusion framework, the denoising dynamics of diffusion models can be cast as a Markov decision process.

$$\mathbf{s}_t \triangleq (\mathbf{c}, t, \mathbf{z}_t), \qquad \pi(\mathbf{a}_t \mid \mathbf{s}_t) \triangleq p(\mathbf{z}_{t-1} \mid \mathbf{z}_t, \mathbf{c}),$$

$$P(\mathbf{s}_{t+1} \mid \mathbf{s}_t, \mathbf{a}_t) \triangleq (\delta_{\mathbf{c}}, \delta_{t-1}, \delta_{\mathbf{z}_{t-1}}), \qquad R(\mathbf{s}_t, \mathbf{a}_t) \triangleq \begin{cases} r(\mathbf{z}_0, \mathbf{c}), & \text{if } t = 0, \\ 0, & \text{otherwise.} \end{cases} \tag{5}$$

In the formulation, $\mathbf{c}$ denotes the action-conditioning input, and $\pi(\mathbf{a}_t \mid \mathbf{s}_t)$ represents the transition probability from the latent state $z_t$ to $z_{t-1}$. Each trajectory consists of $T$ timesteps, after which the transition function $P$ leads to a terminal state. The detailed reward $r(\mathbf{z}_0, \mathbf{c})$ is given in Equation 3. We provide reward only at $t = 0$ for the final output, with no reward at any other timestep.

## 5 EXPERIMENTS

### 5.1 SETUPS

**Implementation Detail**    We use the VPT dataset (Baker et al., 2022) for post-training. We apply a preprocessing pipeline that removes data that cannot be processed by the Inverse Dynamics Model (IDM), specifically frames recorded during GUI interactions or those in which the scene is static. All visual inputs are resized to $384 \times 224$ pixels. In practice, about 1,000 training samples are sufficient for the model to converge.

All training is conducted on AMD MI300X GPUs. In the post-training stage, we load the pretrained weights of MineWorld (Guo et al., 2025b) and NFD (Cheng et al., 2025) for corresponding experiments. For the IDM, we use the VPT-pretrained model (Baker et al., 2022), trained on 2,000 hours of carefully curated gameplay videos. We observe that the prediction accuracy of IDM increases with video length. Therefore, we set the inference length to 16 frames during training. Additional hyperparameter settings are provided in Appendix B.

In evaluation stage, all hyperparameters are kept identical to baseline settings. For MineWorld, we set Top-$p$ sampling to 0.8. For NFD, we use DPM-Solver++ (Lu et al., 2025) with 18 sample steps.

**Evaluation Protocol**    Evaluation proceeds as follows: given an initial frame and a sequence of 15 actions, the model predicts the next frame at each step conditioned on the action associated with the current frame, producing a 16-frame video that we assess for video quality and action following. For video quality, we report Fréchet Video Distance (FVD) (Unterthiner et al., 2018), PSNR (Hore & Ziou, 2010) and VBench (Huang et al., 2024), which measure dynamics and visual quality. For action following, we adopt the MineWorld evaluation protocol and use the IDM to infer actions from videos. We report F1, precision, and recall scores to evaluate the action classification accuracy. We use the official data split from MineWorld (Guo et al., 2025b). To ensure comparability with prior work, we use the results reported in MineWorld and NFD as baselines. More details are listed in Appendix B. We also conduct human evaluations to validate that these metrics align with human preferences.

### 5.2 MAIN RESULTS

As shown in Table 1, the post-trained models significantly outperform the baselines across different paradigms and parameter scales, yielding substantial improvements in action-following accuracy, as reflected by higher F1, recall, and precision scores for actions. Moreover, visual quality metrics such as FVD, PSNR and image quality from VBench also show improvements relative to the baselines. We also report the action-following metric of ground truth videos in the table, which represents the upper bound of the IDM's accuracy and therefore serves as the theoretical upper limit for RLIR performance. After post-training, the model nearly attains this bound.

### 5.3 QUALITATIVE ANALYSIS

Figure 3 presents qualitative comparisons of generations before and after RLIR post-training; additional examples are given in Appendix D. In the first case, the baseline model fails to accurately depict the hand's digging action, yielding a mismatch between excavation progress. In the second case, the baseline shows limited fine-grained distance perception, causing a noticeable misalignment of the character's position with the ground truth. In the final case, under rapid movement, the baseline produces localized pixel blur. By contrast, the RLIR-post-trained model effectively resolves these issues, in line with the improvement in quantitative results.

We also conduct human evaluation as a complement to automatic metrics. For both the autoregressive world model and the diffusion world model, we randomly sample 10 videos from the evaluation set. Evaluators score each output along two dimensions: action-following ability and visual quality. The corresponding ground-truth videos are provided as references for judging action-following. As shown in Figure 4, the models post-trained with RLIR exhibit a clear and consistent improvement over their counterparts on both criteria.

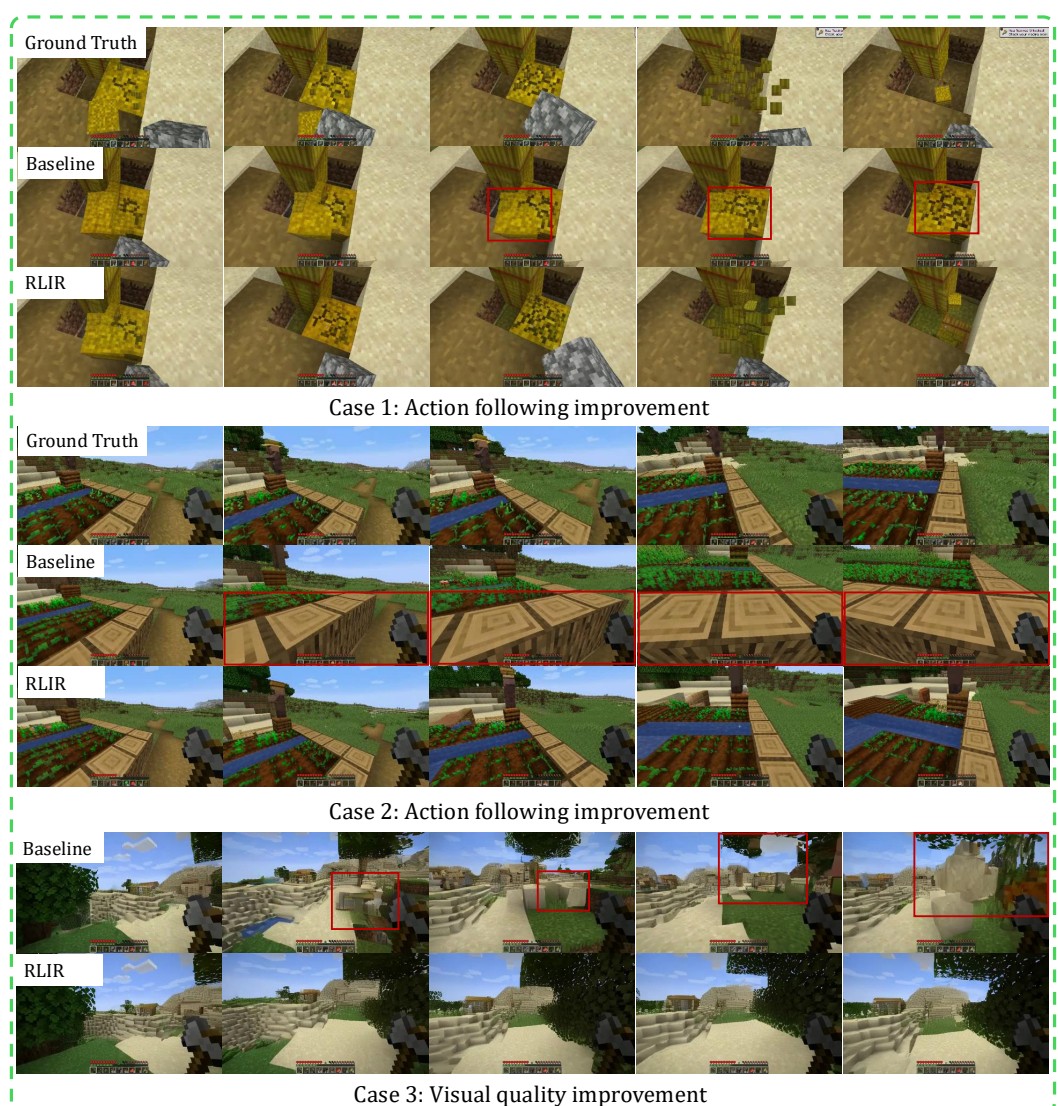

Figure 3: Qualitative comparison between RLIR and baseline. The figure shows the ground truth, the baseline output, and the output after RLIR post-training. No ground truth is required for visual quality cases. The key regions in the image are marked with red boxes. Post-training with RLIR mitigates action inconsistencies and image blurring.

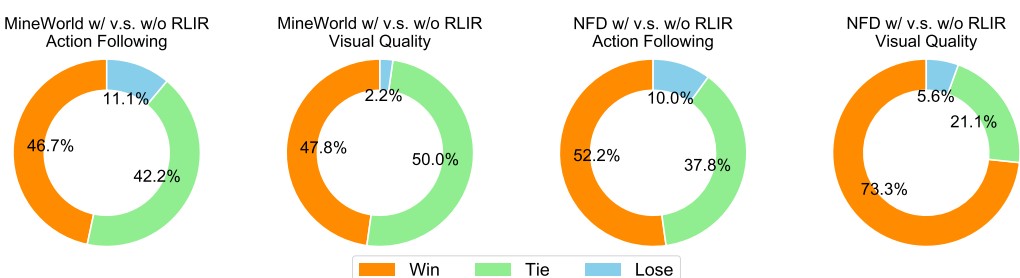

Figure 4: Human evaluation results for MineWorld and NFD with or without RLIR post-training. "Win" indicates the post-training results outperform the original one, while "Lose" represents the opposite. The results demonstrate that RLIR post-training yields higher human preference ratings for both visual quality and action-following ability.

Table 1: Comparison of results with and without RLIR post-training across two model architectures and diverse parameter scales. RLIR consistently improves action-following ability and visual quality. "GT" denotes ground truth videos. "Img. Qual." is short for "image quality".

| Model | Param. | F1↑ | Recall↑ | Precision↑ | FVD↓ | PSNR↑ | Img. Qual.↑ | Dynamic |
|---|---|---|---|---|---|---|---|---|
| **Mine-World** | 300M | 0.70 | 0.71 | 0.72 | 246 | 15.13 | 0.675 | 0.97 |
| | w/ RLIR | 0.77 | 0.76 | 0.79 | 231 | 15.58 | 0.672 | 0.97 |
| | 700M | 0.70 | 0.71 | 0.72 | 231 | 15.32 | 0.677 | 0.96 |
| | w/ RLIR | **0.81** | 0.80 | **0.84** | 207 | 15.78 | 0.678 | 0.97 |
| | 1200M | 0.76 | 0.73 | 0.73 | 227 | 15.69 | 0.682 | 0.97 |
| | w/ RLIR | **0.81** | **0.81** | 0.83 | **205** | **15.99** | **0.684** | 0.96 |
| **NFD** | 310M | 0.69 | 0.69 | 0.71 | 212 | 16.46 | 0.678 | 1.00 |
| | w/ RLIR | 0.76 | 0.76 | 0.77 | 195 | 17.38 | 0.687 | 0.99 |
| | 774M | 0.77 | 0.78 | 0.78 | 184 | 16.95 | **0.692** | 0.99 |
| | w/ RLIR | **0.83** | **0.83** | **0.85** | **180** | **17.48** | 0.688 | 1.00 |
| **GT** | ——— | 0.87 | 0.86 | 0.88 | — | —— | 0.704 | 1.00 |

## 6 ANALYSIS

### 6.1 DIFFERENT REWARD FUNCTIONS

We evaluate the effectiveness of Reinforcement Learning with Inverse Rewards (RLIR) by comparing it with human preference reward (e.g., VideoAlign (Liu et al., 2025b)) and pixel-level verifiable reward proposed in RLVR-World (Wu et al., 2025b). VideoAlign uses 180k human-preference annotations to train a reward model that assigns separate scores to visual quality, motion dynamics, and text alignment. Since the world model is not text-conditioned, we use the mean of the first two dimensions as the reward. In contrast, RLVR-World treats the ground truth video directly as a verifiable reward signal. Concretely, its reward is the sum of the $L_1$ loss and the perceptual loss (LPIPS) between the predictions and the ground-truth frames, $x_i$ means the i-th frame in the video:

$$R_{T_j} = -\sum_{i=1}^{n} \left[ L_1(\hat{x}_i, x_i) + \text{LPIPS}(\hat{x}_i, x_i) \right] \tag{6}$$

We apply both methods to MineWorld and NFD during post-training. The results and reward curves are shown in Table 2 and Figure 5. The ineffectiveness of VideoAlign is straightforward to explain: human evaluators introduce substantial bias and noise, and subtle motion differences in videos are difficult for them to discern. For RLVR-World, we attribute the lack of gains to four main factors:

- **Correlation with pre-training objectives.** The pixel-level supervision provided by $L_1$ + LPIPS is highly correlated with the pre-training objectives (cross-entropy loss for MineWorld or flow matching loss for NFD), which have already been optimized. Consequently, the policy initialization starts near a local optimum, restricting the effective exploration of RL.

- **Uniform weighting of all pixels.** Both $L_1$ and LPIPS aggregate errors over the entire frame, implicitly treating all regions as equally important. For action-following evaluation, however, regions associated with the agent's motion are more critical. In contrast, an Inverse Dynamics Model naturally allocates greater attention to action-relevant regions.

- **Conflict with the generative objective.** In many settings, a world model must synthesize genuinely novel content (e.g., regions uncovered during agent exploration). In such cases, pixel-level rewards are ill-suited, as the newly generated areas lack a deterministic ground truth. The reward should instead emphasize the fidelity of controllable factors (e.g., the magnitude and direction of motion), rather than penalize non-unique viusal outputs.

- **Susceptible to reward hacking.** In our experiments, we find that the videos produced by RLVR-World exhibit an overall dark appearance. This is because part of the post-training dataset

is relatively dark. Therefore, the model can inflate the reward by uniformly darkening frames rather than improving semantic or dynamical fidelity. In contrast, RLIR depends on the predicted action alignment and is largely invariant to global brightness shifts, making it more robust.

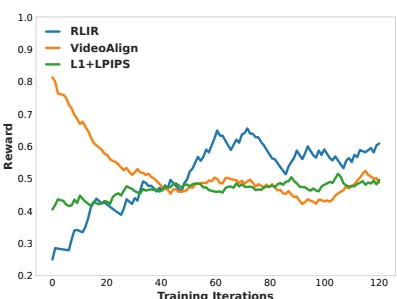

Figure 5: Reward curves for VideoAlign, L1+LPIPS, and RLIR. We rescale the rewards to range (0, 1) for representation.

| Model | Method | F1↑ | FVD↓ | PSNR↑ | IQ↑ |
|---|---|---|---|---|---|
| **Mine-World** | Baseline | 0.70 | 231 | 15.32 | 0.677 |
| | w/ $L_1$ + LPIPS | 0.71 | 228 | 15.47 | 0.673 |
| | w/ VideoAlign | 0.73 | 219 | 15.50 | 0.669 |
| | w/ RLIR | **0.81** | **207** | **15.78** | **0.678** |
| **NFD** | Baseline | 0.77 | 184 | 16.95 | 0.692 |
| | w/ $L_1$ + LPIPS | 0.77 | 193 | 17.09 | 0.645 |
| | w/ VideoAlign | 0.76 | 181 | 17.45 | **0.689** |
| | w/ RLIR | **0.83** | **180** | **17.48** | 0.688 |

Table 2: Performance differences among three methods on 700M-parameter models, IQ is short for "image quality". Pixel-level verifiable reward yields no consistent improvements across models, and the human-preference reward likewise fails to improve performance significantly.

In addition, regarding reward granularity (Razin et al., 2025), the human-preference model provides a coarse, video-level reward, whereas applying RLVR directly yields an overly fine, pixel-level signal. Both extremes are suboptimal. In contrast, RLIR offers a precise, semantically aligned frame-level reward that better matches the training requirements of world models.

## 6.2 ABLATION ON HYPERPARAMETERS

We perform separate ablation studies on the principal hyperparameters of the autoregressive world model and the diffusion world model.

For MineWorld, we evaluate whether adding a KL penalty improves performance. We test across different model sizes and find that the KL penalty yields measurable gains only for small models.

For NFD, we perform ablations on the number of denoising steps and the SDE noise level $\epsilon_t$. When the number of denoising steps increases to 40, performance improves slowly and marginally; the best results occur with 10–20 steps. Setting the noise level too low diminishes gains, while performance remains similar for noise levels between 0.5 and 0.75. Ablation results appear in Appendix C.

## 7 CONCLUSION

We introduce Reinforcement Learning with Inverse Reward (RLIR), a novel framework for world model post-training that transforms unverifiable videos into verifiable rewards. By leveraging the Inverse Dynamics Model (IDM) to convert videos into corresponding action sequences, we are able to measure the performance of the world model by the accuracy of predicted actions. This accuracy is then used as the verifiable reward in the reinforcement learning post-training process to optimize the world model. Experiments conducted on both autoregressive and diffusion world models demonstrate the effectiveness of the proposed method, achieving a 5%–10% improvement in action-following accuracy and enhancing visual quality as well.

**Limitations** (1) As the IDM cannot achieve perfect accuracy, the attainable performance of our method is bounded by the IDM's accuracy. (2) Constrained by computational and data resources, the largest model used in this work has only 1.2 billion parameters. As a result, the base model may fall short of the performance upper bound that RLIR could theoretically achieve.

**Future Work** Future work will evaluate the scalability of RLIR on larger-scale world models and broaden its applications, including other world models and modalities beyond video.

## REPRODUCIBILITY

We provide comprehensive implementation details, including model architectures, training configurations, and codes in Appendix and supplementary materials.

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

# ICLR Paper Appendix for Reinforcement Learning with Inverse Rewards for World Model Post-training

## A  Declaration of LLM Usage

We use large language models (LLMs), including ChatGPT, to support manuscript preparation. Their use are limited to language editing (grammar, spelling, and word choice), code formatting (e.g., adding comments to the code), and drafting figures to aid the creation of final visualizations. All scientific ideas, analyses, and conclusions were conceived, validated, and interpreted independently by the authors. We gratefully acknowledge the assistance of large language models in our work.

## B  More Implementation Details

### B.1  Details of Evaluation Metrics

The Imaging Quality metric in VBench primarily evaluates low-level distortions in generated video frames (e.g., overexposure, noise, blur). VBench uses the MUSIQ Ke et al. (2021) image-quality predictor, which can accommodate variable aspect ratios and resolutions. Each per-frame score (originally in the range 0–100) is divided by 100 to map it to [0,1], and the final metric is the arithmetic mean of the normalized scores across all frames in the video.

For action following metrics, actions in Minecraft can be grouped into 9 classes, where 7 of them represent discrete action classes and the other 2 represent camera movement angles. For discrete classes, each one of them contains two or three exclusive actions such as (`forward`,`backward`) and (`left`, `right`). We provide the full grouping results in Table 3. Then, by taking the provided action as the ground truth and the predicted action from IDM as the prediction, we can utilize

commonly utilized classification metrics including precision, recall and F1 score to evaluate the classification accuracy. We report both the macro scores to reduce the effect of imbalanced labels.

Table 3: Classification Tasks and Their Labels

| Task Type | Actions | Labels |
|---|---|---|
| Triple Classification | `forward`, `backward`
`left`, `right`
`sprint`, `sneak` | `forward`, `backward`, `null`
`left`, `right`, `null`
`sprint`, `sneak`, `null` |
| Binary Classification | `use`
`attack`
`jump`
`drop` | `use`, `null`
`attack`, `null`
`jump`, `null`
`drop`, `null` |

## B.2 MODEL CONFIGURATONS

**MineWorld**  We apply the proposed algorithm and post-train three MineWorld models of different sizes—300M, 700M, and 1.2B parameters—based on the LLaMA architecture. The base model configurations are summarized in Table 4.

Table 4: The configuration of different size of MineWorld models.

| | Hidden Dim. | MLP Dim. | Num. Heads | Num. Layers |
|---|---|---|---|---|
| 300M | 1024 | 4096 | 16 | 20 |
| 700M | 2048 | 4096 | 32 | 20 |
| 1.2B | 2048 | 8192 | 32 | 20 |

**NFD**  We post train on 300M and 770M parameter NFD models. Their base configurations are summarized in Table 5. The NFD architecture comprises Diffusion Transformer Blocks.

NFD employs an image-level tokenizer to transform each frame into a sequence of latents to enable the frame-level interaction with the model. For actions, NFD quantizes camera angles into discrete bins, and categorize other actions into 7 exclusive classes, each represented by a unique token.

NFD leverages a Block-wise Causal Attention mechanism that combines bidirectional attention within each frame and causal dependencies across frames to model spatio-temporal dependencies efficiently. For each token in a frame, it will attend to all tokens within the same frame (i.e., intra-frame attention), as well as to all tokens in preceding frames (i.e., causal inter-frame attention).

NFD utilizes a linear layer to map the actions into action vectors and adopt adaLN-zero conditioning.

Table 5: The configuration of different size of NFD models.

| | Hidden Dim. | MLP Dim. | Num. Heads | Num. Layers |
|---|---|---|---|---|
| 310M | 1024 | 2730 | 16 | 16 |
| 774M | 1536 | 4096 | 24 | 18 |

## B.3 EXPERIMENTAL SETTINGS

**MineWorld**  Table 6 lists the hyperparameters used for MineWorld post-training.

**NFD**  Table 7 summarizes the hyperparameters used for NFD post-training.

Table 6: Hyper-parameters for MineWorld.

| Hyperparameter | Value |
|---|---|
| Learning rate scheduler | cosine |
| Learning rate | $3e^{-5}$ |
| Optimizer | AdamW |
| Rollout | 24 |
| Clip Ratio | 0.2 |
| Samples per iteration | 32 |

Table 7: Hyper-parameters for NFD.

| Hyperparameter | Value |
|---|---|
| Learning rate scheduler | cosine |
| Learning rate | $1e^{-5}$ |
| Optimizer | AdamW |
| Rollout | 24 |
| Clip Ratio | 0.2 |
| Samples per iteration | 16 |
| Sampling steps | 10 |
| Noise level $\epsilon_t$ | 0.75 |
| Timestep Selection $\tau$ | 0.6 |

## C  ABLATION STUDIES

### C.1  MINEWORLD

For the autoregressive world models, we investigate the effect of introducing a KL-divergence constraint. The KL penalty term is used to regulate the divergence between the online policy and the frozen reference policy, the goal of KL-divergence is to align the model behavior without diverging too far from the initial model. With a fixed KL penalty of 1e-4 across model sizes, smaller models benefit (better action following and visual quality), whereas larger models suffer performance drops. We show the effectiveness of the KL penalty across all models in Table 8.

Table 8: Ablation study on kl penalty. the kl penalty coefficient is set to $\beta = 1e - 4$.

| Model | F1↑ | Recall↑ | Precision↑ | FVD↓ | PSNR↑ | Img. Qual.↑ | Dynamic |
|---|---|---|---|---|---|---|---|
| 300M w/ $kl$ | 0.77 | 0.76 | 0.79 | 231 | 15.58 | 0.672 | 0.97 |
| 300M w/o $kl$ | 0.69 | 0.69 | 0.73 | 231 | 15.65 | 0.672 | 0.98 |
| 700M w/ $kl$ | 0.73 | 0.73 | 0.75 | 210 | 15.91 | 0.678 | 0.98 |
| 700M w/o $kl$ | 0.81 | 0.80 | 0.84 | 207 | 15.78 | 0.678 | 0.97 |
| 1200M w/ $kl$ | 0.80 | 0.79 | 0.82 | 219 | 15.80 | 0.683 | 0.97 |
| 1200M w/o $kl$ | 0.81 | 0.81 | 0.83 | 205 | 15.99 | 0.684 | 0.96 |

### C.2  NFD

To investigate the impact of different timesteps on optimization, we keep other hyperparameters constant and test 10, 20, and 40 steps. When the number of denoising steps is increased to 40, performance improved only slowly and marginally; the best results were observed with 10–20 steps. Regarding noise level, we test 0.25, 0.5 and 0.75. Setting a too low noise level diminishes performance

gains, while results remain similar for noise levels between 0.5 and 0.75. Table 9 and 10 shows the effect of denoising steps and noise level.

Table 9: Ablation study of NFD on denoising steps.

| Model | Step | F1↑ | Recall↑ | Precision↑ | FVD↓ | PSNR↑ | Img. Qual.↑ | Dynamic |
|-------|------|-----|---------|-----------|------|-------|-------------|---------|
| **310M** | 10 | 0.76 | 0.76 | 0.77 | 195 | 17.38 | 0.687 | 0.99 |
| | 20 | 0.74 | 0.74 | 0.76 | 186 | 17.23 | 0.687 | 1.00 |
| | 40 | 0.70 | 0.70 | 0.74 | 221 | 16.72 | 0.667 | 1.00 |
| **774M** | 10 | 0.83 | 0.83 | 0.85 | 180 | 17.48 | 0.688 | 1.00 |
| | 20 | 0.81 | 0.84 | 0.80 | 185 | 17.35 | 0.683 | 1.00 |
| | 40 | 0.77 | 0.78 | 0.78 | 180 | 17.47 | 0.686 | 1.00 |

Table 10: Ablation study of NFD on noise level.

| Model | Noise Level ($\epsilon_t$) | F1↑ | Recall↑ | Precision↑ | FVD↓ | PSNR↑ | Img. Qual.↑ | Dynamic |
|-------|-------|-----|---------|-----------|------|-------|-------------|---------|
| **310M** | 0.75 | 0.76 | 0.76 | 0.77 | 195 | 17.38 | 0.687 | 0.99 |
| | 0.50 | 0.75 | 0.75 | 0.76 | 196 | 17.39 | 0.684 | 0.99 |
| | 0.25 | 0.76 | 0.76 | 0.77 | 199 | 17.35 | 0.687 | 0.99 |
| **774M** | 0.75 | 0.83 | 0.83 | 0.85 | 180 | 17.48 | 0.688 | 1.00 |
| | 0.50 | 0.83 | 0.83 | 0.84 | 187 | 17.40 | 0.683 | 0.99 |
| | 0.25 | 0.83 | 0.83 | 0.84 | 183 | 17.43 | 0.684 | 1.00 |

## D  MORE VISUALIZATION RESULTS

We present additional visualization cases in Figure 6: the first three cases are from NFD, and the last case is from MineWorld. More videos can be found in the supplementary material.

## E  REBUTTAL

### E.1  CLARIFICATION ON METHODOLOGICAL NOVELTY

As mentioned in our manuscript, our core contribution lies in **employing an inverse model to map inherently unverifiable video outputs into verifiable, low-dimensional action sequences**. Our method has undergone extensive testing across different model architectures and various scales, demonstrating strong empirical gains. We would like to highlight that these contributions have been recognized by the reviewers:

- **Novelty:** Reviewer xznu has explicitly acknowledged the innovation of our proposed method.
- **Significance:** Reviewers 3FNX and pmFp considers the problem we address to be interesting and crucial.
- **Performance:** Reviewers xznu, pmFp, and ZYxS have validated the outstanding performance and effectiveness of RLIR.

### E.2  JUSTIFICATION FOR THE MINECRAFT ENVIRONMENT

Our work's primary focus is to validate the efficacy of inverse rewards for world model RL. To this end, we evaluated our method on both diffusion-based and autoregressive-based world models. We selected Minecraft as our evaluation environment for the following reasons:

- **Diverse Visual Space:** Minecraft features a comprehensive visual space, which is analogous to complex real-world domains such as autonomous driving or robotics, making it an ideal testbed for visual generalization.

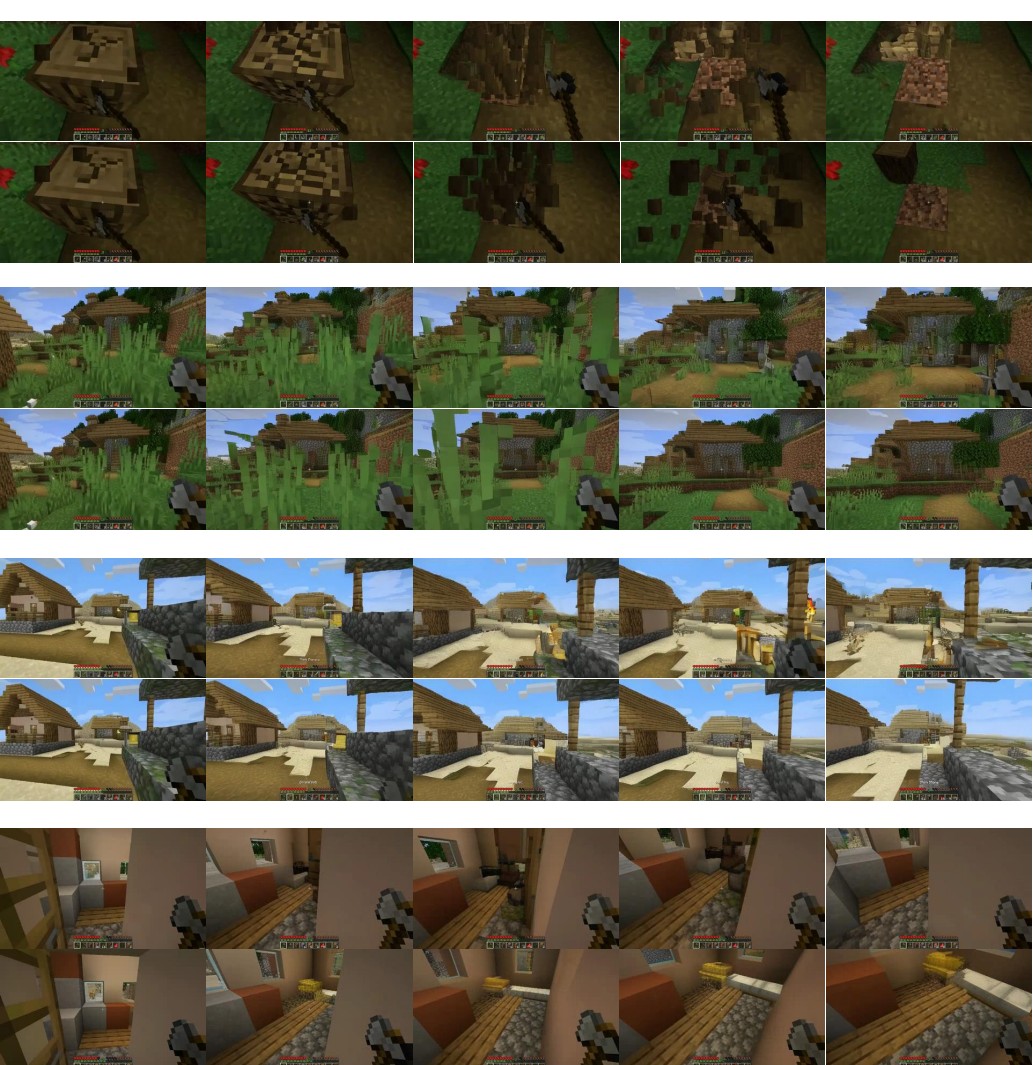

Figure 6: More Qualitative results. The top row displays the baseline output, and the bottom row is the output after post-training.

- **Complex Action Space:** The VPT dataset used in our experiments contains a complex action space. This is comparable to and arguably more complex than those found in other well-known world models (e.g., Genie3 (Google, 2025), Mirage2 (team, 2025)). Consequently, we believe it is highly representative in terms of action space complexity.
- **Resource and Data Constraints:** In alternative domains, there is currently a lack of sufficient labeled data, or the prohibitive GPU resources required to train a world model from scratch are unavailable.

The robust performance achieved in both settings corroborates the effectiveness of our approach. Should other open-source world models become available in the future, we would be delighted to extend our evaluation to include them.

### E.3 ADDITIONAL EXPERIMENTS: DEMONSTRATING THE GAP BETWEEN SFT AND RL

To further investigate the performance disparity between SFT and RL, we trained world models using Supervised Fine-Tuning (SFT) on the VPT dataset for 1,000 steps, utilizing ground truth labels for model optimization. The corresponding results are presented below.

Table 11: Performance of Autoregressive World Model with SFT and RLIR.

| Param. | Method | F1↑ | Recall↑ | Precision↑ | FVD↓ | PSNR↑ | Img. Qual.↑ | Dynamic |
|---|---|---|---|---|---|---|---|---|
| 300M | RLIR | 0.77 | 0.76 | 0.79 | 231 | 15.58 | 0.672 | 0.97 |
| | SFT | 0.72 | 0.72 | 0.74 | 242 | 15.21 | 0.670 | 0.97 |
| 700M | RLIR | 0.81 | 0.80 | 0.84 | 207 | 15.78 | 0.678 | 0.97 |
| | SFT | 0.70 | 0.71 | 0.72 | 232 | 15.44 | 0.673 | 0.96 |
| 1200M | RLIR | 0.81 | 0.81 | 0.83 | 205 | 15.99 | 0.684 | 0.97 |
| | SFT | 0.77 | 0.76 | 0.78 | 229 | 15.78 | 0.680 | 0.97 |

Table 12: Performance of Diffusion World Model with SFT and RLIR.

| Param. | Method | F1↑ | Recall↑ | Precision↑ | FVD↓ | PSNR↑ | Img. Qual.↑ | Dynamic |
|---|---|---|---|---|---|---|---|---|
| 310M | RLIR | 0.76 | 0.76 | 0.77 | 195 | 17.38 | 0.687 | 0.99 |
| | SFT | 0.69 | 0.69 | 0.72 | 208 | 17.01 | 0.679 | 0.98 |
| 774M | RLIR | 0.83 | 0.83 | 0.85 | 180 | 17.48 | 0.688 | 1.00 |
| | SFT | 0.76 | 0.76 | 0.78 | 181 | 17.51 | 0.690 | 0.99 |

Our experiments indicate that RL is indeed necessary. We hypothesize that the observed performance advantage of Reinforcement Learning (RL) over Supervised Fine-Tuning (SFT) is primarily attributed to their distinct optimization mechanisms.

When using ground truth labels for SFT, the model is limited to learning exclusively from 'positive' samples. Consequently, actions that are not well-learned or are performed sub-optimally cannot be sufficiently improved through SFT alone.

In contrast, our RLIR framework addresses this directly: Actions that are already executed well produce minimal to zero gradients (as their advantage approaches zero when the group consensus is high). Conversely, actions that are difficult to perform correctly receive a much larger gradient signal, as determined by the normalized Advantage:

$$A_i = \frac{r_i - \text{mean}\left(\{ri\}_{i=1}^{G}\right)}{\text{std}\left(\{r_i\}_{i=1}^{G}\right)}, \tag{7}$$

where $G$ represent the group size. Therefore, this mechanism allows the model to focus its optimization efforts precisely on the most challenging actions, leading to more effective and targeted improvement.

### E.4 Ablation Study: Impact of IDM Accuracy on Performance

We investigated the sensitivity of RLIRto the quality of the Inverse Dynamics Model (IDM). To simulate IDMs of varying accuracy levels in a controlled manner, we stochastically inverted the original IDM's predictions with noise probabilities of 25%, 50%, and 75%. While this simulation is not strictly equivalent to training an IDM with naturally lower accuracy, this approach allows for a more precise and quantifiable evaluation of noise tolerance. The results are presented below.

Table 13: Ablation study regarding the inversion ratio on both Autoregressive and Diffusion models.

| Param. | Method | F1↑ | Recall↑ | Precision↑ | FVD↓ | PSNR↑ | Img. Qual.↑ | Dynamic |
|--------|--------|-----|---------|-----------|------|-------|-------------|---------|
| | | *Autoregressive World Model* | | | | | | |
| | RLIR | 0.81 | 0.81 | 0.83 | 205 | 15.99 | 0.684 | 0.97 |
| **1200M** | 25% invert | 0.77 | 0.78 | 0.78 | 214 | 15.68 | 0.678 | 0.96 |
| | 50% invert | 0.69 | 0.69 | 0.72 | 228 | 15.77 | 0.684 | 0.97 |
| | 75% invert | 0.72 | 0.72 | 0.74 | 231 | 15.34 | 0.684 | 0.97 |
| | | *Diffusion World Model* | | | | | | |
| | RLIR | 0.83 | 0.83 | 0.85 | 180 | 17.48 | 0.688 | 1.00 |
| **774M** | 25% invert | 0.78 | 0.79 | 0.79 | 185 | 16.62 | 0.688 | 0.99 |
| | 50% invert | 0.76 | 0.76 | 0.77 | 191 | 16.85 | 0.682 | 1.00 |
| | 75% invert | 0.75 | 0.75 | 0.76 | 187 | 16.43 | 0.690 | 0.99 |

When the IDM accuracy dropped to 50% or below, the action-following ability of the world model showed no improvement, and reward cannot increase during training. This indicates the resulting reward signal becomes meaningless. The performance observed from the 75% accuracy model and the 90.6% (original) accuracy model confirms that a higher-quality IDM directly contributes to a better-performing world model.

Whether the world model can be successfully optimized is directly related to the inherent capabilities of the world model itself. Therefore, we empirically believe that a more accurate IDM will yield greater performance improvements.

### E.5 Clarification on Action Space and Action-Following Definitions

We would like to clarify the distinction between **Action Following** and **Temporal Consistency**, as they represent fundamentally different evaluation dimensions:

- **Temporal Consistency** primarily measures the visual coherence and smoothness of the video sequence. As seen in benchmarks like VBench (Huang et al., 2024), this is often quantified by computing the CLIP feature similarity between consecutive frames.

- **Action Following** requires that the generated next frame must accurately reflect the specific conditioning action.

To illustrate the difference, consider a scenario where the input action is 'move left'. A generated video that smoothly depicts a 'move right' trajectory might still achieve a high Temporal Consistency score due to visual coherence. However, for Action Following, this same video would receive a score of 0, as it directly contradicts the control signal. Therefore, Action Following measures *controllability* and *correctness*, whereas Temporal Consistency measures *visual smoothness* regardless of the condition.

**Details on Action Space.** Regarding the specific definition of our action space, the action recorded per frame originates in a raw format, containing detailed mouse and keyboard events (e.g., `{"mouse":{"x":872.0, ...}, "keyboard":{"keys":["w", "space"]}}`). Following the standard established in VPT (Baker et al., 2022), we convert this raw input into a structured, discretized representation (e.g., `{"forward": 1, "jump": 1, "camera": [10, 9]}`), which subsequently serves as the conditioning input for our model.

### E.6 MARKOV DECISION PROCESS DEFINITION FOR AUTOREGRESSIVE WORLD MODEL

The MDP definitions of Diffusion World Model have been provided in Section 4.3. We formulate the Autoregressive World Model generation process as a Markov Decision Process (MDP) defined by the tuple $(\mathcal{S}, \mathcal{A}, \pi, P)$:

- **State Space ($\mathcal{S}$):** The state $s_t$ is the sequence of all tokens processed by the model up to time $t$. This sequence comprises the input context (initial historical frame tokens and input action tokens) and all previously generated image tokens.

- **Action Space ($\mathcal{A}$):** The action $a_t$ is a discrete token selected from the model's output vocabulary, specifically corresponding to the set of valid image tokens.

- **Policy ($\pi$):** The policy is the trained autoregressive model itself, denoted as $\pi_\theta(a_t | s_t)$, which defines the probability distribution for the next image token $a_t$ given the current state $s_t$.

- **Transition Dynamics ($P$):** The transition dynamics are deterministic but conditional on frame completion. Let $N_t$ denote the number of image tokens generated for the current frame within $s_t$ (where a full frame consists of 336 tokens). Using $\oplus$ to represent concatenation, the state transition is defined as:

$$s_{t+1} = \begin{cases} s_t \oplus a_t & \text{if } (N_t \bmod 336) \neq 0 \\ s_t \oplus a_t \oplus \mathbf{c}_{\text{next}} & \text{if } (N_t \bmod 336) = 0 \end{cases} \tag{8}$$

where $\mathbf{c}_{\text{next}}$ represents the *next-action-tokens* required to initiate the generation of the subsequent frame.

