# OpenReview forum: "Reinforcement Learning with Inverse Rewards for World Model Post-training"
_ICLR.cc/2026/Conference — Submitted to ICLR 2026_

### Official Review · Reviewer_ZYxS · 2025-10-29

**Soundness:** 3
**Presentation:** 3
**Contribution:** 2
**Rating:** 4
**Confidence:** 3

**Summary:**

This paper proposes RLIR (Reinforcement Learning with Inverse Rewards), a post-training framework for action-conditioned world model designed to improve action-following ability using an inverse dynamics model (IDM). Rather than relying on human annotation or hand-crafted video verifiers, RLIR maps generated videos to the action space by inferring actions from the predicted future states via IDM, comparing against ground-truth actions to create a verifiable reward for GRPO-based post-training. The method is evaluated on both autoregressive and diffusion-based world models. Experimental results demonstrate consistent improvements over baselines in action-following accuracy, visual quality and human preference metrics.

**Strengths:**

-Efficient Action-Consistent Rewarding: The core idea, using an IDM to verify whether the world model outputs actually encode the given actions, is a practical, scalable alternative to expensive or brittle human/hand-crafted reward schemes.
-Comprehensive Experimentation: Experimental results span autoregressive (MineWorld) and diffusion-based (NFD) architectures across model sizes; Tables 1–2 show consistent gains (~5–10% action-following) alongside improved visual quality.
-Rigorous Baseline and Ablation: The paper rigorously benchmarks against alternate reward shaping baselines (VideoAlign, RLVR-World pixel-level rewards), with ablation on reward function properties, hyperparameters (Table 8-10), and action/verifiability structure.
-Substantive Qualitative Demonstration: Figure 3 (Page 7) provides explicit, step-by-step visual evidence: RLIR-corrected models address fine-grained failures in the baseline, such as action misalignment and localized visual artifacts, supporting key claims on both metrics.
-Plug-and-play compatibility: RLIR is architecture-agnostic and integrates with GRPO without modifying the backbone, suggesting easy transfer across model families.

**Weaknesses:**

-Data leakage risk: What worries me most is if the IDM and world model share training distributions, the reward may encode dataset priors rather than transferable action understanding.
-Training details and reproducibility gaps: The proposed method relies on an Inverse Dynamics Model but provides no reproducible training details—data/labeling, architecture, objective, hyperparameters, or validation. This undermines reproducibility and confidence in the reward/credit assignment.
-Dependence on IDM Fidelity: As noted in the limitations (p. 9), IDM errors cap reward quality. The paper does not quantify boundary conditions or adversarial/ambiguous cases where IDM mistakes could mis-reward or mis-penalize the model; robustness analysis is missing.
-Generality Across Domains: Validation is limited to Minecraft-style environments with a strong pre-trained IDM (VPT). It is unclear how RLIR/IDM extend to open-ended, multimodal, or real-world domains with continuous or ambiguous actions.
-Interpretability and Failure Cases: While qualitative improvements are highlighted (Figure 3), there is little diagnosis of RLIR’s failure modes. For example, are there patterns in which RLIR models overfit the IDM or fail to improve when IDM is already near-saturated? Do there exist situations where the world model “hacks” the IDM in a manner akin to classic reward hacking phenomena?
-Scalability Limitations: The largest model is 1.2B parameters (Table 1). Costs, verifier throughput, and latency trade-offs at larger scales remain unclear.
-Unproven planning benefit: Although RLIR post-trains world models, there is no controlled evidence that the resulting models improve planning or RL (closed-loop task performance, sample efficiency, or long-horizon generalization).

**Questions:**

-Data availability: Can you introduce the IDM training dataset (or a reproducible subset) and the RLIR post-training dataset/splits (including hashes, preprocessing, and licensing) to enable replication.
-IDM Robustness and Generalization: Can you provide a detailed evaluation of IDM failure cases, especially in adversarial or ambiguous visual/action states? How do you prevent the RLIR-trained world model from hacking IDM?
-Out-of-Domain application/scalability: What modifications are required for continuous or multimodal action spaces, or partially observable settings? Any pilot evidence?
-Planning/RL evidence: Can you provide closed-loop planning/MBRL results (task success, sample efficiency, long-horizon returns) with ablations against a no-RLIR baseline?
-Human preference details: Please report rater instructions, sampling, and inter-rater agreement to ensure robustness.
-Potential action Imbalance: Does the IDM training dataset exhibit action-distribution imbalance, and if so, how is it addressed (e.g., reweighting, resampling, per-class metrics)?
Rating: 6, marginally above the acceptance threshold. But would not mind if paper is rejected.

---

> ### Author Response · Authors · 2025-11-20
> **Rebuttal by Authors**
>
> Thank you for the time, thorough comments, and nice suggestions. We believe the most of concerns have already presented in our original paper. We are also slightly puzzled that your rating does not seem to align with the comment in the last line of the "Questions" section. We have summarized your questions and will answer them point by point.
>
> > **Q1:** Data leakage risk for IDM & IDM Robustness & prevent hacking IDM & IDM reproducibility & IDM Fidelity & IDM training dataset exhibit action-distribution imbalance
>
> **A1:** Our experiments demonstrate that the IDM is capable of highly accurate action prediction. The IDM can achieve  90.6% accuracy on keypress prediction and an R^2 of 0.97 for mouse movement regression, which we also mentioned **in Section 3**. This aligns with the findings in Video PreTraining (VPT) [1].
>
> We did not observe any reward hacking in our pretrained IDM. Both a reduction in image quality and incorrect actions lead to a decrease in the IDM's prediction accuracy. Therefore, the model is optimized towards performing more accurate actions while also maintaining or improving image quality. We also discuss why reward hacking does not occur **in Section 6**.
>
> The IDM used is pre-trained. We have thoroughly detailed the IDM's implementation **in Section 3.1** of the paper. The IDM architecture is comprised primarily of a temporal convolution layer, a ResNet image processing stack, and residual unmasked attention layers, which more details can be found in [1].
>
> We utilize a pre-trained IDM, and thus our method does not involve the IDM training process, so we do not have the issue of "action imbalance" in IDM training.
>
> ---
>
> > **Q2:** Data availability
>
> **A2:** We utilized an open-source dataset: VPT[1], and the details of the data processing are outlined **in Section 5.1**. We apply a preprocessing pipeline that removes data that cannot be processed by the Inverse Dynamics Model, specifically frames recorded during GUI interactions or those in which the scene is static. All visual inputs are resized to 384 × 224 pixels.
>
>
> ---
>
> > **Q3:** Robustness analysis
>
> **A3:** We evaluated RLIR on both diffusion-based and autoregressive-based world models and different model sizes. We have also included an extensive analysis of critical hyper-parameters in the Appendix. Our experimental results indicate that our method is insensitive to hyper-parameters. Furthermore, we observe significant improvements across both diffusion-based and autoregressive world models of varying parameter sizes, demonstrating the robustness of our approach. We hope our results can addresses your concerns. Additionally, both Reviewer 3FNX and Reviewer pmFp acknowledged our framework and experiments.
>
> ---
> > **Q4:** Generality Across Domains
>
> **A4:** We evaluated RLIR on Minecraft for the following reasons:
> - **Representative benchmark for video world model:** Minecraft data has a comprehensive visual space, which is analogous to other domains like autonomous driving or robotics.
> - **Diverse and complex action space:** The VPT dataset on Minecraft that we used has a diverse action space, which is comparable to, and arguably more complex than those of other well-known world models (e.g., Genie3 [1], Mirage2 [2]). Consequently, we believe it is already highly representative in terms of action space complexity.
> - **Resource and Data Constraints:** In alternative domains, we lack access to sufficient labeled data and the substantial GPU resources required to train a world model from scratch.
>
> Our work's primary focus is to validate the efficacy of inverse rewards for world model RL. To this end, we evaluated our method on both diffusion-based and autoregressive-based world models. The strong performance achieved in both settings corroborates the effectiveness of our approach. If other open-source world models become available in the future, we would be delighted to test our methodology on them.
>
> ---
> > **Q5:** Planning/RL evidence
>
> **A5:** The objective of our work is to enhance the world model's action following capability; it does not involve a planning process.
>
> ---
> > **Q6:** Human preference details
>
> **A6:** The details of our human evaluation are described **in Section 5.3** of the paper. For both the autoregressive world model and the diffusion world model, we randomly sample 10 videos from the evaluation set. Evaluators score each output along two dimensions: action-following ability and visual quality. The corresponding ground-truth videos are provided as references for judging action-following. Additionally, our human evaluation set is included in the supplementary materials.
>
>
> ---
>
> [1] Baker B, Akkaya I, Zhokov P, et al. Video pretraining (vpt): Learning to act by watching unlabeled online videos[J]. Advances in Neural Information Processing Systems, 2022, 35: 24639-24654.
>
> [2] Google. Genie 3. https://deepmind.google/discover/blog/genie-3-a-new-frontier-for-world-models/.
>
> [3] Mirage 2. https://www.mirage2.org.

---

> ### Comment · Reviewer_ZYxS · 2025-11-26
> **Office Comment by Reviewer ZYxS**
>
> Thanks a lot for the author’s detailed response. Firstly, I apologize for the confusion caused by the score in the question. This is due to incorrect pasting. The author has satisfactorily addressed most of my questions regarding the Inverse Dynamics Model (IDM) and the experimental details. Besides, I would like to put forward two remaining suggestions for the authors' consideration:
>
> 1.	Data Issues: If I understand correctly, IDM training (described in VPT), Backbone training (described in Mineworld), world model post-training, and evaluation all occur on VPT. However, the paper's description of training data only includes a brief introduction in Section 5.1. I recommend that the author clarify the split of these training data segments to make it clearer.
>
> 2.	Cross-dataset validation: Although the authors claim that the dataset used in the paper is representative, validation on other datasets (e.g., robotic manipulation) with different action types or visual styles would still help make the conclusion of the work more convincing.
>
> Based on these points, I am inclined to maintain my original score.

---

### Official Review · Reviewer_pmFp · 2025-10-30

**Soundness:** 2
**Presentation:** 2
**Contribution:** 2
**Rating:** 4
**Confidence:** 2

**Summary:**

This paper proposes a verifiable reward for world model post-training via a pretrained IDM. This reward makes it verifiable whether frames generated actually carry out the action inputs. GRPO with this reward shows better generation compared to human preference reward and pixel-level L1 reward, and both diffusion and autoregressive models can benefit from this reward.

**Strengths:**

1. The subject being studied is interesting. Whether the action is correctly carried out is an angle from which the understanding capability of the world model could be strengthen.

2. Both autoregressive and diffusion based world models can benefit from the method proposed. Empirical results beat two baseline reward functions.

**Weaknesses:**

1. The definition of "Action Following" is vague. Literally, it's hard to distinguish it from temporal consistency where both of them refer to the generated frames contain unwilling behaviors, based on which I would say action following is a subproblem of temporal consistency. This makes it hard for me to understand the visualization in Fig.3 where authors claim that action following is improved.

2. The method proposed lacks novelty to me. It seems like simply apply VPT to serve as supervision, despite RL regime it uses such as GRPO on both autoregressive and diffusion models.

3. This method is heavily dependent on the quality of pretrained IDM, which poses a limitation on those tasks without a large-scale pretrain dataset from which a precise IDM can be trained.

**Questions:**

1. Related to W3,  is there any cases of reward hacking observed due to an unprecise estimate by IDM?

2. Suppose IDM provides us precise action-label estimates, what are the results if we SFT finetune the WM on that IDM-annotated dataset similarly using action following distance as the SFT loss? Is RL necessary for improving action following?

---

> ### Author Response · Authors · 2025-11-20
> **Rebuttal by Authors-1**
>
> Thanks for your time and the constructive suggestions. Here are additional responses and clarifications based on your comments.
>
>
> > **Q1:** The definition of "Action Following" is vague.
>
> **A1:** We would like to clarify the distinction. Action Following and Temporal Consistency are different concepts. Action Following requires that the generated next frame must **accurately reflect the specific input action**. Temporal Consistency merely ensures that the video frames are visually coherent and smooth sequentially. To be specific, as seen in benchmarks like VBench [1], temporal consistency is often measured by computing the CLIP feature similarity between consecutive frames. This metric primarily captures visual smoothness.
>
> Consider a scenario where the input action is 'move left':
> A generated video that depicts 'moving right' might still achieve a high Temporal Consistency score, as the frames themselves are coherent.
>
> However, for Action Following, **this same video would receive a score of 0, as it directly contradicts the given action.** Therefore, "Action Following" measures correctness based on the action-condition, while "Temporal Consistency" measures smoothness regardless of the condition.
>
> Reviewer 3FNX similarly acknowledges the crucial nature of action following for controllable world models.
>
> ---
>
> > **Q2:** The method relies heavily on the quality of the pretrained IDM, limiting its use on tasks lacking large-scale pretraining data.
>
> **A2:** We argue that training an Interaction Dynamics Model (IDM) is often less demanding than training a full world model. This is because the IDM is only required to predict low-dimensional actions, which are discrete in many scenarios.
>
> For instance, in the Anypos [2], a collection of merely 610k image-action pairs was sufficient to enhance the IDM's prediction accuracy by 51%. In contrast, 610k image pairs are demonstrably insufficient for training a precise video world model (e.g., ImageNet contains over 1 million examples). This is because the IDM is required to predict low-dimensional actions from high-dimensional video inputs, while video world model generates high-dimensional video from low-dimensional actions.
>
> Given the same amount of data, an IDM is significantly easier to train than a world model, and can therefore consistently provide guidance for it. While the IDM's accuracy in certain contexts is not yet optimal, we believe this approach offers a more scalable and data-efficient research direction.
>
> ---
> > **Q3:** Is there any cases of reward hacking observed due to an unprecise estimate by IDM?
>
> **A3:** We did not observe any reward hacking in our pretrained IDM. Both a reduction in image quality and incorrect actions lead to a decrease in the IDM's prediction accuracy. Therefore, the model is optimized towards performing more accurate actions while also maintaining or improving image quality. We also discuss why reward hacking does not occur in Section 6.
>
>
>
>
>
> ---
>
> [1] Huang, Ziqi, et al. "Vbench: Comprehensive benchmark suite for video generative models." Proceedings of the IEEE/CVF Conference on Computer Vision and Pattern Recognition. 2024.
>
> [2] Tan, Hengkai, et al. "Anypos: Automated task-agnostic actions for bimanual manipulation." arXiv preprint arXiv:2507.12768 (2025).

---

> ### Author Response · Authors · 2025-11-20
> **Rebuttal by Authors-2**
>
> > **Q4:** Explain the novelty of RLIR. and what are the results if we SFT finetune the WM on annotated dataset? Is RL necessary for improving action following?
>
> **A4:**
> Our main contributions are listed below:
> - we are the first to identify and address the action following problem in world model, a contribution explicitly acknowledged by Reviewer 3FNX.
> - in tackling this novel setting, we identified a critical challenge: the video modality historically lacks verifiable rewards. To overcome this, we propose the RLIR algorithm. Reviewer xznu also identified this as a strength of our paper.
> - we validate the effectiveness of this reward formulation on both autoregressive and diffusion models. Our comparisons with direct SFT further demonstrate that adopting RL provides meaningful improvements.
>
> We train world models using Supervised Fine-Tuning(SFT) on VPT datasets for 1000 steps, which utilize ground truth labels for model optimization. The results are listed below.
>
> **Autoregressive World Model**
> | Method | Param. | F1 | Recall | Precision | FVD | PSNR | Img. Qual. | Dynamic |
> | :--- | :---: | :---: | :---: | :---: | :---: | :---: | :---: | :---: |
> |  RLIR  |300M |      0.77    |    0.76    |    0.79    |    231    |    15.58  |     0.672    |      0.97   |
> |  SFT   |300M |      0.72    |    0.72    |    0.74    |    242    |    15.21  |     0.670    |      0.97   |
> |  RLIR  |700M |      0.81    |    0.80    |   0.84     |    207    |    15.78  |     0.678    |      0.97   |
> |  SFT   |700M |      0.70    |    0.71    |    0.72    |    232    |    15.44  |     0.673    |      0.96   |
> |  RLIR  |1200M |      0.81    |    0.81    |   0.83     |    205    |    15.99  |     0.684    |      0.97   |
> |  SFT   |1200M |      0.77    |    0.76    |    0.78    |    229    |    15.78  |     0.680    |      0.97   |
>
>
>
> **Diffusion World Model**
> | Method | Param. | F1 | Recall | Precision | FVD | PSNR | Img. Qual. | Dynamic |
> | :--- | :---: | :---: | :---: | :---: | :---: | :---: | :---: | :---: |
> |  RLIR  |310M |     0.76    |    0.76    |    0.77    |    195    |    17.38   |    0.687     |      0.99   |
> |  SFT   |310M |      0.69    |    0.69    |    0.72    |    208    |    17.01  |     0.679    |      0.98   |
> |  RLIR  |774M |      0.83    |    0.83    |   0.85     |    180    |    17.48  |     0.688    |     1.00    |
> |  SFT   |774M |     0.76   |    0.76    |    0.78    |    181    |    17.51  |     0.690    |      0.99   |
>
> Our experiments indicate that RL is indeed necessary. We hypothesize that the observed performance advantage of Reinforcement Learning (RL) over Supervised Fine-Tuning (SFT) is primarily attributed to the following mechanisms: When using ground truth labels for SFT, the model is limited to learning exclusively from 'positive' samples. Consequently, actions that are not well-learned or are performed sub-optimally cannot be sufficiently improved through SFT. In our RLIR framework, this is addressed directly: Actions that are already executed well produce minimal to zero gradients (as their Advantage is zero when all actions in the group are correct). Actions that are difficult to perform correctly receive a much larger gradient signal, as determined by the normalized Advantage (Advantage = (r_i - mean) / std). Therefore, this mechanism allows the model to focus its optimization efforts precisely on the most challenging actions, leading to more effective and targeted improvement.

---

> > ### Comment · Reviewer_pmFp · 2025-11-26
> >
> > Thank you for your response, yet it has not fully addressed my concerns. You mentioned AnyPos is able to train on 610k dataset, but it only achieves a 57.13% accuracy on test set, which is hardly sufficient for serving as action following supervision. As for your setting, VPT includes 100 hours of videos, accounting for about 72M frames given a frame at 20fps, and this is only for a low-data regime while a ~1900 hours dataset is used for best performance. Moreover, as Reviewer 3FNX also puts it, the novelty of the method proposed, still, is not convinced to me. So I will keep my assessment unchanged and maintain my original score.

---

### Official Review · Reviewer_3FNX · 2025-11-01

**Soundness:** 3
**Presentation:** 3
**Contribution:** 2
**Rating:** 4
**Confidence:** 4

**Summary:**

This paper addresses inaccurate action-following in video world models. It proposes RLIR, a post-training framework that uses an Inverse Dynamics Model (IDM) to recover actions from generated videos, converting high-dimensional video outputs into low-dimensional, verifiable rewards. The GRPO algorithm optimizes the world model. Experiments on MineWorld and NFD models with VPT videos show 5–10% improvements in action-following, visual quality, and higher human preference scores. The paper introduces RLIR for post-training, validates it across autoregressive and diffusion paradigms, and demonstrates consistent gains.

**Strengths:**

1. The paper addresses a key gap: While prior world models focus on visual quality, this paper targets action-following, which is crucial for controllable world models. The IDM-based rewards avoid costly human annotations or rule-based verifiers.
2. The paper clearly describes IDM training, GRPO optimization, and adaptations for autoregressive and diffusion models, demonstrating a coherent and well-structured framework.

**Weaknesses:**

1. Leveraging an IDM for reward construction is conceptually similar to curiosity-driven RL approaches that use inverse models to guide exploration. While RLIR applies this concept for post-training world models with an action-alignment reward, the paper does not discuss these prior connections, leaving the core insight only moderately novel.
2. The paper lacks a clear definition of the MDP used in RL optimization. Key components—state space (e.g., historical frames, latent representations), action space (mapping generative steps to RL actions), and transition dynamics—are not explicitly described. This ambiguity makes it difficult to fully understand how RL interacts with world model generation.
3. Experiments are conducted solely on the VPT dataset (Minecraft gameplay), which restricts conclusions about RLIR’s generalization. Validation on other domains (e.g., robotic manipulation, autonomous driving, real-world scene simulation) with different action types or visual scenarios would strengthen claims of general applicability.

**Questions:**

1. GRPO is claimed to be more memory-efficient than PPO by removing the value network. However, large-scale world models (1B+ parameters) have high computational demands. Could you provide quantitative data on training time and memory overhead of RLIR post-training compared to baseline fine-tuning methods (e.g., supervised action-label fine-tuning)? This would clarify RLIR’s practicality at scale.
2. RL is proposed as necessary for optimizing world models with IDM-derived rewards, but why not use direct gradient propagation through IDM? For example, treating IDM’s action prediction loss as a supervision signal for end-to-end fine-tuning. Could you include a comparative experiment between RLIR and direct end-to-end fine-tuning, or explain why the latter is infeasible (e.g., instability, conflicting objectives)?
3. The IDM’s accuracy bounds RLIR’s performance, yet only a pre-trained VPT IDM is used. Have you investigated how IDM accuracy impacts RLIR results? For instance, ablating IDM with different prediction accuracies or fine-tuning IDM on subsets of VPT to reduce accuracy. This would clarify RLIR’s sensitivity to IDM quality and the potential gains from improving IDM further.

---

> ### Author Response · Authors · 2025-11-20
> **Rebuttal by Authors-1**
>
> Thanks for your time and the constructive suggestions. Here are additional responses and clarifications based on your comments.
>
> > **Q1:** The paper's IDM-based reward is conceptually similar to curiosity-driven RL, but it overlooks these connections, making the insight only moderately novel.
>
> **A1:** We would like to clarify that our core contribution is **the definition and validation of inverse action as a verifiable reward within a world model**, which we believe is fundamentally different from curiosity-driven RL.
>
> We have revisited "Curiosity-driven Exploration by Self-supervised Prediction" [1], which we believe the reviewer is referring to. Curiosity-driven RL formulates an intrinsic reward as the prediction error between the predicted and actual state representations in a visual feature space learned via a self-supervised inverse dynamics model, thereby encouraging the agent to explore novel states where its internal model is currently inaccurate.
>
> In curiosity-driven RL, given image frames, the agent explores to discover better action strategies. Curiosity-driven RL aims at making agent explore environments more effectively in sparse reward. Specifically, the model is optimized by minimizing the error between the IDM-predicted action and the actual action. **In contrast, our work utilizes the predicted action to provide supervision in the image dimension.** Secondly, the inverse dynamics model is not frozen in curiosity-driven RL, while in RLIR, **the IDM is frozen during training.**
>
> Both the reward definition, application tasks, purpose and the implementation of the inverse dynamics model bear little resemblance to our proposed method. Therefore, we respectfully maintain that this comparison is not directly relevant to our work. Nevertheless, we find your point to be highly insightful and are more than willing to incorporate it into the Related Work section.
>
> ---
>
> > **Q2:** The paper fails to clearly define the MDP components (state, action, transitions) for RL, making its interaction with world model generation difficult to understand.
>
> **A2:** Regarding the diffusion world model, we have provided the MDP definitions in **Section 4.3** of the manuscript. For the autoregressive world model, we provide the explicit MDP components below and will ensure the correct details are added to the revised manuscript after discussion:
>
> - State Space ($S$): The state $s_t$ is the sequence of all tokens processed by the model up to time $t$. This sequence includes the initial historical frame tokens and input action tokens (i.e., the input context) and all previously generated image tokens.
>
> - Action Space ($A$): The action $a_t$ is a single token selected from the model's output vocabulary, which specifically corresponds to the set of all possible image tokens.
>
> - Policy ($\pi$): The policy is the trained autoregressive model itself, $\pi_\theta(a_t | s_t)$. It defines the probability distribution for selecting the next image token $a_t$ given the current state $s_t$.
>
> - Transition Dynamics ($P$): The transition dynamics $P(s_{t+1} | s_t, a_t)$ are deterministic. Following the standard LLM RL definition, the next state $s_{t+1}$ is formed by concatenating the current state $s_t$ with the chosen action $a_t$. However, the transition is conditional on whether the current frame is complete. Once a frame is complete, we should concatenate next action tokens for next frame generation. In our implementation, one image frame requires 336 tokens. Let *len(image tokens)* be the number of image tokens generated for the current frame (as contained in $s_t$), and $\oplus$ denote concatenation.The dynamics are defined as follows:
>
>     $ s_{t+1} = s_t \oplus a_t, \quad \text{if (len(image tokens)\\%336)} \neq 0 $
>
>     $ s_{t+1} = s_t \oplus a_t \oplus \text{next-action-tokens}, \quad \text{if (len(image tokens)\\%336)} = 0 $
>
> ---
>
> [1] Pathak, Deepak, et al. "Curiosity-driven exploration by self-supervised prediction." International conference on machine learning. PMLR, 2017.

---

> ### Author Response · Authors · 2025-11-20
> **Rebuttal by Authors-2**
>
> > **Q3:** Validation on other domains would strengthen claims of general applicability.
>
> **A3:** Our work's primary focus is to validate the efficacy of inverse rewards for world model RL. To this end, we evaluated our method on both diffusion-based and autoregressive-based world models. We evaluated RLIR on Minecraft for the following reasons:
> - **Representative benchmark for video world model:** Minecraft data has a comprehensive visual space, which is analogous to other domains like autonomous driving or robotics.
> - **Diverse and complex action space:** The VPT dataset on Minecraft that we used has a diverse action space, which is comparable to, and arguably more complex than those of other well-known world models (e.g., Genie3 [1], Mirage2 [2]). Consequently, we believe it is already highly representative in terms of action space complexity.
> - **Resource and Data Constraints:** In alternative domains, we lack access to sufficient labeled data and the substantial GPU resources required to train a world model from scratch.
>
> The strong performance achieved in both settings corroborates the effectiveness of our approach. If other open-source world models become available in the future, we would be delighted to test our methodology on them.
>
> ---
>
> > **Q4:**  Could you provide quantitative data on training time and memory overhead of RLIR post-training compared to baseline fine-tuning methods?
>
>
> **A4:** We provide the following results on training time and memory overhead.
>
>
> **Training Time**
>
> Our method requires only 90 steps (Autoregressive model) or 80-120 steps (Diffusion model) to see significant effects.
> - For 1.2B AR Model (90 steps): ~16 hours on 8x Mi300x. Due to environment challenge and time limitation, we did not adapt vllm to accelerate the rollout process.
>
> - 774M Diffusion Model (120 steps): ~1 hour.
> The main time bottleneck is the rollout (inference) phase, not the RL update. IDM inference time is negligible.
>
> **Memory Overhead**
>
> - Rollout Phase: Memory consumption is identical to standard world model inference, which is not the bottleneck.
> - Gradient Update Phase: We use GRPO, which already removes the value network. Compared to a standard GRPO baseline, our only additional cost is a single, frozen 482M Inverse Dynamics Model (~0.9GB in FP16).
>
> Since the IDM is lightweight and frozen (no gradients or optimizer states), we conclude that RLIR's practicality and scaling are comparable to a standard GRPO implementation.
>
> ---
> [1] Google. Genie 3. https://deepmind.google/discover/blog/genie-3-a-new-frontier-for-world-models/.
>
> [2] Mirage 2. https://www.mirage2.org.

---

> ### Author Response · Authors · 2025-11-20
> **Rebuttal by Authors-3**
>
> > **Q5:** why is direct gradient propagation infeasible? Please provide a comparative experiment between RLIR and direct end-to-end fine-tuning, or explain why the latter is impractical.
>
> **A5:** We train world models using Supervised Fine-Tuning(SFT) on VPT datasets for 1000 steps, which utilize ground truth labels for model optimization. The results are listed below.
>
> **Autoregressive World Model**
> | Method | Param. | F1 | Recall | Precision | FVD | PSNR | Img. Qual. | Dynamic |
> | :--- | :---: | :---: | :---: | :---: | :---: | :---: | :---: | :---: |
> |  RLIR  |300M |      0.77    |    0.76    |    0.79    |    231    |    15.58  |     0.672    |      0.97   |
> |  SFT   |300M |      0.72    |    0.72    |    0.74    |    242    |    15.21  |     0.670    |      0.97   |
> |  RLIR  |700M |      0.81    |    0.80    |   0.84     |    207    |    15.78  |     0.678    |      0.97   |
> |  SFT   |700M |      0.70    |    0.71    |    0.72    |    232    |    15.44  |     0.673    |      0.96   |
> |  RLIR  |1200M |      0.81    |    0.81    |   0.83     |    205    |    15.99  |     0.684    |      0.97   |
> |  SFT   |1200M |      0.77    |    0.76    |    0.78    |    229    |    15.78  |     0.680    |      0.97   |
>
>
>
> **Diffusion World Model**
> | Method | Param. | F1 | Recall | Precision | FVD | PSNR | Img. Qual. | Dynamic |
> | :--- | :---: | :---: | :---: | :---: | :---: | :---: | :---: | :---: |
> |  RLIR  |310M |     0.76    |    0.76    |    0.77    |    195    |    17.38   |    0.687     |      0.99   |
> |  SFT   |310M |      0.69    |    0.69    |    0.72    |    208    |    17.01  |     0.679    |      0.98   |
> |  RLIR  |774M |      0.83    |    0.83    |   0.85     |    180    |    17.48  |     0.688    |     1.00    |
> |  SFT   |774M |     0.76   |    0.76    |    0.78    |    181    |    17.51  |     0.690    |      0.99   |
>
>
>
> Our experiments indicate that RL is indeed necessary. We hypothesize that the observed performance advantage of Reinforcement Learning (RL) over Supervised Fine-Tuning (SFT) is primarily attributed to the following mechanisms: When using ground truth labels for SFT, the model is limited to learning exclusively from 'positive' samples. Consequently, actions that are not well-learned or are performed sub-optimally cannot be sufficiently improved through SFT. In our RLIR framework, this is addressed directly: Actions that are already executed well produce minimal to zero gradients (as their Advantage is zero when all actions in the group are correct). Actions that are difficult to perform correctly receive a much larger gradient signal, as determined by the normalized Advantage (Advantage = (r_i - mean) / std). Therefore, this mechanism allows the model to focus its optimization efforts precisely on the most challenging actions, leading to more effective and targeted improvement.

---

> ### Author Response · Authors · 2025-11-21
> **Rebuttal by Authors-4**
>
> > **Q6:** IDM accuracy bounds RLIR performance; only a pre-trained IDM was used. Did you investigate how IDM accuracy affects RLIR results (e.g., by ablating/fine-tuning IDM)? This would clarify RLIR's sensitivity to IDM quality.
>
>
>
> **A6:** We followed your suggestion to investigate this. We simulated IDMs of lower quality by **stochastically inverting the original IDM's predicted accuracy with probabilities of 25%, 50%, and 75%**. Although this ablation experiment is not strictly equivalent to directly reducing the IDM's accuracy, this approach provides better quantification. The results are as follows:
>
> **Autoregressive World Model**
>
>
> | Model | Param. | F1 | Recall | Precision | FVD | PSNR | Img. Qual. | Dynamic |
> | :--- | :---: | :---: | :---: | :---: | :---: | :---: | :---: | :---: |
> |  RLIR  |1200M |      0.81    |    0.81    |   0.83     |    205    |    15.99  |     0.684    |      0.97   |
> |  25% invert |1200M |      0.77    |    0.78    |   0.78     |    214    |    15.68  |     0.678    |      0.96   |
> |  50% invert  |1200M |      0.69    |    0.69    |   0.72     |    228    |    15.77  |     0.684    |      0.97   |
> |  75% invert  |1200M |      0.72    |    0.72    |   0.74     |    231    |    15.34  |     0.684    |      0.97   |
>
> **Diffusion World Model**
>
> | Model | Param. | F1 | Recall | Precision | FVD | PSNR | Img. Qual. | Dynamic |
> | :--- | :---: | :---: | :---: | :---: | :---: | :---: | :---: | :---: |
> |  RLIR  | 774M |      0.83    |    0.83    |   0.85     |    180    |    17.48  |     0.688    |      1.00   |
> |  25% invert | 774M |      0.78    |    0.79    |   0.79     |    185    |    16.62  |     0.688    |      0.99   |
> |  50% invert  | 774M |      0.76    |    0.76    |   0.77     |    191    |    16.85  |     0.682    |      1.00   |
> |  75% invert  | 774M |      0.75    |    0.75    |   0.76     |    187    |    16.43  |     0.690    |      0.99   |
>
> When the IDM accuracy dropped to 50% or below, the action-following ability of the world model showed no improvement, and reward cannot increase during training. This indicates the resulting reward signal becomes meaningless. The performance observed from the 75% accuracy model and the 90.6% (original) accuracy model confirms that a higher-quality IDM directly contributes to a better-performing world model.
>
> Whether the world model can be successfully optimized is directly related to the inherent capabilities of the world model itself. Therefore, we empirically believe that a more accurate IDM will yield greater performance improvements. A more accurate IDM will yield greater performance improvements.

---

> > ### Comment · Reviewer_3FNX · 2025-11-22
> >
> > Thank you for the response. However, the current version of the paper still does not fully convince me in terms of experimental sufficiency and methodological novelty. I have therefore decided to maintain my original score. In addition, I suggest that the authors clearly highlight the revised parts in the updated manuscript (though this is not related to my score decision).

---

> > > ### Author Response · Authors · 2025-11-23
> > >
> > > Thank you for your response. As suggested, we have included the revised content in Appendix E of the manuscript. Regarding your remaining concerns, we would like to offer the following clarifications:
> > >
> > > **On Experimental Sufficiency:** We have implemented the additional experiments requested in your initial review and have included the detailed results in Appendix E of the revised manuscript. If there are specific remaining gaps or additional baselines you consider essential, please let us know, and we will be happy to address them.
> > >
> > > **On Methodological Novelty:** As elaborated in our previous response, our work addresses the critical and under-explored challenge of "action following" in world models. We would like to highlight that the problem setting and the proposed IDM-based reward design were recognized as key strengths by Reviewers xznu, pmFp, ZYxS.
> > >
> > > We are fully open to further discussion and hope these clarifications help address your concerns.

---

### Official Review · Reviewer_xznu · 2025-11-07

**Soundness:** 3
**Presentation:** 4
**Contribution:** 2
**Rating:** 6
**Confidence:** 2

**Summary:**

The authors present post-training method that improves how well video world models follow input actions by using an Inverse Dynamics Model to turn generated videos back into predicted actions and reward the model based on their agreement. This avoids costly human preferences or hand-crafted video verifiers.

**Strengths:**

- new reward design avoids need for human preference labels or video checkers.
- re-uses pre-trained IDMs
- strong empirical gains

**Weaknesses:**

- limited evaluation outside of minecraft.
- better description of the minecraft action space is required. unclear if authors are using the full keyboard + mouse action space or the simplified action space.

**Questions:**

- do the authors have alternative datasets they have evaluated their method on?

---

> ### Author Response · Authors · 2025-11-20
> **Rebuttal by Authors**
>
> Thank you for your valuable feedback! Your endorsement of our paper gives us significant encouragement. Here are our clarifications.
>
>
> > **Q1:** Could the authors detail any evaluations conducted on alternative datasets outside of the Minecraft environment?
>
> **A1:**
> Our work's primary focus is to validate the efficacy of inverse rewards for world model RL. To this end, we evaluated our method on both diffusion-based and autoregressive-based world models. We evaluated RLIR on Minecraft for the following reasons:
> - **Representative benchmark for video world model:** Minecraft data has a comprehensive visual space, which is analogous to other domains like autonomous driving or robotics.
> - **Diverse and complex action space:** The VPT dataset on Minecraft that we used has a diverse action space, which is comparable to, and arguably more complex than those of other well-known world models (e.g., Genie3 [1], Mirage2 [2]). Consequently, we believe it is already highly representative in terms of action space complexity.
> - **Resource and Data Constraints:** In alternative domains, we lack access to sufficient labeled data and the substantial GPU resources required to train a world model from scratch.
>
> The strong performance achieved in both settings corroborates the effectiveness of our approach. If other open-source world models become available in the future, we would be delighted to test our methodology on them.
>
> ---
>
> > **Q2:** better description of the Minecraft action space is required. unclear if authors are using the full keyboard + mouse action space or the simplified action space.
>
> **A2:** In the standard Minecraft setting, raw keyboard and mouse inputs are remapped to game-defined camera movements and actions. Thus, our action space is fully consistent with the game's standard control definitions.
>
> More specifically, the action recorded per frame is in a raw format. For example:
> {"mouse":{"x":872.0,"y":327.0,"scaledX":232.0,"scaledY":-33.0},"keyboard":{"keys":["key.keyboard.w","key.keyboard.space"]}}.
> Following the standard established in VPT [3], this raw input is then converted into a structured representation (e.g., {"forward": 1, "jump": 1, "camera": [10, 9]}), which subsequently serves as the conditioning input for our model.
>
>
> ---
>
> [1] Google. Genie 3. https://deepmind.google/discover/blog/genie-3-a-new-frontier-for-world-models/.
>
> [2] Mirage 2. https://www.mirage2.org.
>
> [3] Baker B, Akkaya I, Zhokov P, et al. Video pretraining (vpt): Learning to act by watching unlabeled online videos[J]. Advances in Neural Information Processing Systems, 2022, 35: 24639-24654.

---

### Author Response · Authors · 2025-11-23
**General Response**

We sincerely thank the reviewers for their insightful and constructive feedback. All reviewers (xznu, 3FNX, pmFp, ZYxS) acknowledged the value of our method in addressing the critical gap of action consistency for controllable world models. They appreciated the efficiency and novelty of our IDM-based reward design (xznu, 3FNX, ZYxS) and highlighted the consistent empirical gains achieved across both autoregressive and diffusion-based architectures (xznu, pmFp, ZYxS). Additionally, the rigorous experimental analysis and the clarity of the presentation were commended (3FNX, ZYxS).

In response to these comments, we have addressed the common concerns in Appendix E of the revised manuscript. We also provide point-by-point responses to each reviewer. The major updates include:

- Clarification of Novelty: We have expanded the discussion on the novelty of our method.

- Environment Justification: We have added a detailed explanation for choosing Minecraft as the experimental environment.

- Comparison with SFT: We have included additional experiments comparing our method directly with SFT across both architectures, along with an analysis of the necessity of RL.

- IDM Sensitivity Analysis: We have added ablation studies to analyze the impact of IDM accuracy on model performance.

- Definitions: We have elaborated on the definitions of the action space and "action following."

- MDP Formulation: We have provided a formal definition of the MDP within the RLIR framework.

We are happy to answer any further questions and look forward to the continued discussion.

---

### Meta-Review · Area_Chair_Kpyv · 2026-01-07

**Summary:**

This paper aims to improve the action-following capability in video world models through a post-training framework, Reinforcement Learning with Inverse Rewards (RLIR). The RLIR framework pre-trains an Inverse Dynamics Model (IDM) to recover low-dimensional actions from high-dimensional generated videos into verifiable rewards. The GRPO algorithm is leveraged to post-train both autoregressive and diffusion world models based on the IDM rewards. Empirical evaluation shows some positive performances in action-following, visual quality, and human preference.

The paper has received very insightful, constructive and detailed reviews, and the authors have performed hard work to implement the rebuttal. After rebuttal, reviewers actively participated into the discussion phase, and recognized that several concerns were addressed properly. However, based on the reviewers' final comments, there are still critical aspects outstanding, covering weaknesses in experimental sufficiency, methodological novelty, and cross-dataset generalization. Therefore, the paper still needs a major revision.

**Reviewer Concerns:**

The AC concurs with the final comments raised by the reviewers. The paper falls below the acceptance threshold due to weaknesses in the experimental sufficiency, methodological novelty, and cross-dataset generalization. Specifically:
- Experimental sufficiency: Experiments are conducted solely on the VPT dataset (Minecraft gameplay), which restricts conclusions about RLIR’s generalization to other domains. As per the paper's setting, VPT includes 100 hours of videos only for a low-data regime that falls far behind a ~1900 hours dataset for best performance.
- Methodological novelty: The proposed IDM does not constitute novel designs in model architectures or training methods. The whole solution follows standard practices of RL post-training pipeline.

**Reviewer Scores:**

Reviewers have acknowledged in their final comments that their scores won't be changed.

---

### Decision · Program_Chairs · 2026-01-26

Reject